# SpectralWords: Spectral Embeddings Approach to Word Similarity Task for Large Vocabularies

**Ivan Lobov**
Criteo Research
Paris, France
`i.lobov@criteo.com`

## Abstract

In this paper we show how recent advances in spectral clustering using Bethe Hessian operator (Saade et al., 2014) can be used to learn dense word representations. We propose an algorithm SpectralWords that achieves comparable to the state-of-the-art performance on word similarity tasks for medium-size vocabularies and can be superior for datasets with larger vocabularies.

## 1 Introduction

Learning dense vector representations have been successful in multiple domains including natural language processing for tasks like semantic and syntactic similarities (Levy & Goldberg, 2014), parsing (Chen & Manning, 2014) and translation (Zou et al., 2013).

The family of methods was proposed to build dense representations from naturally occurring texts, most of them explicitly or implicitly learning from co-occurrence counts of pairs of words: Skip-gram with Negative Sampling (SGNS) (Mikolov et al., 2013), Glove (Pennington et al., 2014), SVD on positive PMI (PPMI) matrix (Levy & Goldberg, 2014), Swivel (Shazeer et al., 2016) and others. Recent advances (Qiu et al., 2017) in understanding SGNS suggested a deep connection to spectral methods that use eigenvectors of linear operators based on the adjacency matrix. The classical spectral approaches are known to perform poorly on large, sparse, and high-degree random graphs (Zhang et al., 2012). In our work we explore if approaches like SGNS and SVD on PPMI, similarly, get worse results on larger vocabularies and propose a spectral embedding approach based on the Bethe Hessian operator, which was shown to perform well on sparse graphs (Saade et al., 2014).

The main contribution of our work is a novel approach based on spectral methods leveraging the Bethe Hessian operator. On medium-size vocabularies it achieves comparable results to SGNS and SVD on PPMI on word similarity tasks, and we show that SpectralWords can outperform those methods on larger vocabularies.

## 2 Non-backtracking operator

Classical spectral methods (Von Luxburg, 2007) are based on the properties of leading eigenvectors of adjacency matrix $A$ of an undirected weighted graph $G = (V, E)$ where $V$ is a set of vertices, $E$ is a set of edges and $w_{ij}$ is the weight of the edge between nodes $i$ and $j$. The first eigenvector of $A$ essentially sorts nodes by their degrees (sum of weights of edges incident on the node). The second eigenvector can be used to determine minimum cut on the graph, that is split its nodes into two clusters with the minimum weight of edges connecting two clusters. We call these eigenvectors related to the global structure of the graph *useful*.

Although this approach can be efficient in some cases, it is known to fail for large sparse graphs with high-degree nodes. The phenomena has been well understood for graphs generated by the stochastic block model (SBM). This model assumes $q$ clusters of nodes, the edges are generated randomly from a $q \times q$ matrix of probabilities and it is often assumed to have more edges connecting nodes within the same community than nodes from different communities.

In the case of dense and relatively small graphs generated from SBM, useful eigenvectors are well separated in the spectrum: they are are larger than eigenvalues corresponding to the noise that stay

within $[-2\sqrt{c}, 2\sqrt{c}]$ range where $c$ is the average degree of the graph. But with the growth of the number of nodes while keeping $c$ constant, it's been shown (Krivelevich & Sudakov, 2003) that useful eigenvalues can get mixed in the noisy part of the spectrum dominated by high-degree nodes and hence cannot be efficiently recovered. Other classical operators used in spectral methods, like normalized laplacian or random walk matrix, are prone to the same weakness as mentioned in (Krzakala et al., 2013).

In recent work (Krzakala et al., 2013) it was proposed to use non-backtracking operator $B$ due to its useful properties: when the size of the graph increases its useful eigenvalues stay outside of $[-\sqrt{\rho(B)}, \sqrt{\rho(B)}]$ circle, where $\rho(B)$ is the largest eigenvalue of $B$:

$$B_{i \to j, k \to l} = \delta_{jk}(1 - \delta_{il})w_{ij}$$

Where $\delta_{ij} = 1$ if $i = j$ and $\delta_{ij} = 0$ if $i \neq j$. This operator effectively transforms the initial graph into one that doesn't allow for a random walker to get back to the node that it just came from. Thus this operator has different spectral properties compared to adjacency matrix. Unfortunately, $B$ is of size $2|E| \times 2|E|$, which makes it unfeasible to use even for medium size graphs. However (Saade et al., 2014) proposed a Bethe Hessian operator which is still $|V| \times |V|$, but has the same spectrum and properties of the useful eigenvectors as $B$. It's defined for weighted graphs as follows:

$$\tilde{H}(r)_{ij} = \delta_{ij}\left(1 + \sum_{k \in \Gamma(i)} \frac{A_{ik}^2}{r^2 - A_{ik}^2}\right) - \frac{rA_{ij}}{r^2 - A_{ij}^2} \tag{1}$$

Where $r = \sqrt{\rho(B)}$, $\Gamma(i)$ is a set of neighbors of node $i$ and $A_{ij}$ is a value $ij$ in an adjacency matrix. In our work we use this operator to retrieve useful eigenvectors from the word pair co-occurrence matrix constructed from the training corpus.

## 3 APPROACH

Similar to other word embedding approaches, we consider counts of pairs of words that occurred together within a given window in the sentence in order to define similarity between two words. We found it useful to scale the counts by a factor $\alpha \leq 1$, so we define the adjacency matrix as follows:

$$A_{ij} = \begin{cases} w_{ij}^{\alpha} \text{ if } ij \in E \\ 0 \text{ if } ij \notin E \end{cases}$$

Where $w_{ij}$ is the count of co-occurrences of words $i$ and $j$ within a given window. We then estimate the $\rho(B)$ of the non-backtracking operator by the approximation proposed in (Saade et al., 2014):

$$\hat{\rho}(B) = \sum_i^n d_i^2 / \sum_i^n d_i - 1$$

Where $d_i$ is the sum of weights of edges incident on node $i$. Although this is not the theoretically correct way to estimate largest eigenvalues of $B$ for weighted graphs (see Saade (2016) for the full description of the weighted case), we found that in practice it is a good approximation for graphs based on word co-occurrences.

At the final stage we find the $k$ smallest real eigenvalues and corresponding eigenvectors of Bethe Hessian defined in (1). We use rows of the matrix of stacked eigenvectors as word representations.

## 4 EXPERIMENTS

We collected pair counts from the Wikipedia dump of August 2013 [1]. The corpus contains 77.5 million sentences with 1.5 billion tokens. The preprocessing is done in similar way as in (Levy & Goldberg, 2014). The evaluation tasks are the following: WordSim353 (Finkelstein et al., 2001) partitioned into WordSim Similarity and WordSim Relatedness (Zesch et al., 2008), (Agirre et al., 2009); MEN dataset (Bruni et al., 2012); Mechanical Turk (Radinsky et al., 2011); Rare Words (Luong et al., 2013); and SimLex-999 dataset (Hill et al., 2015).

---

[1]We kindly thank Omer Levy for providing us with the dataset, since it wasn't available online anymore

Table 1: Robustness of embedding quality with growing vocabulary size. The metric is median relative performance decrease in the method performance on similarity tasks, less decrease is better.

| Method | Vocabulary size | | | |
|---|---|---|---|---|
| | $4.4 \times 10^5$ | $9.5 \times 10^5$ | $2.3 \times 10^6$ | $7.7 \times 10^6$ |
| SGNS | $-1\%$ | $-3\%$ | $-33\%$ | $-98\%$ |
| SVD | $-4\%$ | $-12\%$ | $-38\%$ | $-77\%$ |
| SpectralWords | $0\%$ | $-3\%$ | $-21\%$ | $-72\%$ |

Table 2: Performance of Spectral Words vs other methods on similarity tasks. The metric reported is Spearman's correlation, larger scores are better. The confidence intervals reported for SpectralWords are estimated using 100 bootstraps, the intervals are estimates of 95 and 5 percentiles.

| Method | WordSim Similarity | WordSim Relatedness | Bruni et al. MEN | Radinsky et al. M. Turk | Luong et al. Rare Words | Hill et al. SimLex |
|---|---|---|---|---|---|---|
| Glove | 0.725 | 0.604 | 0.729 | 0.632 | 0.403 | **0.398** |
| SGNS | **0.793** | **0.685** | **0.774** | **0.693** | **0.470** | 0.438 |
| SVD | **0.793** | **0.691** | **0.778** | 0.666 | **0.514** | 0.432 |
| SpectralWords | **0.800 ± 0.05** | **0.678 ± 0.07** | **0.782 ± 0.01** | 0.599 ± 0.08 | **0.508 ± 0.05** | **0.471 ± 0.05** |

**SpectralWords learning**[2]    Our method has a single hyperparameter $\alpha$, which was chosen based on grid search with validation on the performance of similarity tasks. For all experiments referenced here we used $\alpha = 0.3$. The partial eigenvalue decomposition is done with Krylov-Schur method (Stewart, 2002) and relative eigenvalue tolerance of $1e-2$.

**Robustness with large vocabularies**    Since approaches like SGNS and SVD on PPMI are based on a form of factorization of a transformed adjacency matrix, we expect them to be prone to the same weakness as other adjacency matrix based methods, namely, the quality of learned embeddings should decrease for matrices of large and sparse graphs. On the other hand, our approach should benefit from the fact that the non-backtracking operator provides better separation of the useful eigenpairs for sparse graphs.

To test this hypothesis, we trained models with fixed hyperparameters (for SGNS and SVD we use those recommended by Levy et al. (2015)), dimension of 100 (due to memory constraints) while decreasing the minimum count threshold for words. With lower threshold the vocabulary is larger and thus the task is harder because underlying graph is significantly more sparse.

We conduct experiments with the following count thresholds $[100, 30, 10, 3, 1]$, report the decrease in embeddings quality compared to the smallest vocabulary based on count threshold of 100 and show results in Table 1. We observe that, as expected, as the vocabulary size grows, all methods performance degrade, but our proposed method is more robust to the increase in the vocabulary size.

**Embeddings quality comparison**    We compare our method to SGNS, SVD and Glove using the results from (Levy & Goldberg, 2014). All experiments are done using the vocabulary of 189,533 words that occurred more than 100 times in the dataset. We train our method with the same window size and number of dimensions as other methods, see the results in Table 2. SpectralWords performs on par with SGNS and SVD on 5 out 6 datasets and outperforms Glove on 4 out of 6 tasks.

## 5    CONCLUSION

We presented a novel approach based on recent advances in spectral methods that is performing better for large and sparse graphs than approaches based on PMI-matrix factorization. Regarding the future work, it would be useful to scale the presented approach for larger datasets and vocabulary sizes and test it for tasks like node classification (Perozzi et al., 2014) or product similarities (Grbovic et al., 2015) that often deal with extremely large vocabularies.

---

[2]We plan on releasing the code to our approach and the Wikipedia dump used for our experiments

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
