# OpenReview forum: "SpectralWords: Spectral Embeddings Approach to Word Similarity Task for Large Vocabularies"
_ICLR.cc/2018/Workshop — Accept_

### Official Review · AnonReviewer1 · 2018-02-28
**Nice insight, method and results**

**Rating:** 7
**Confidence:** 4

**Review:**

This work provides a novel approach to inducing semantic similarity from co-occurrence matrices (harvested from text corpora). The method is based on a recent improved spectral clustering algorithm.

Pros
- The paper is very well written
- The method is clearly explained
- The choice of experiments and comparisons is good for understanding the differences between other approaches

Cons
- The high-level intuition (if there is one) behind the spectral technique (and why it improves similarity) is not made clear.
- It is not clear whether code is made available for this method (highly recommended), or how easy it is to apply.

---

### Official Review · AnonReviewer3 · 2018-03-06
**application of Bethe Hessian to learning word embeddings**

**Rating:** 5
**Confidence:** 4

**Review:**

The paper applies the Bethe Hessian (Saade et al., 2014) to learning word embededings. The technique is interesting and has connections to the non-backtracking walk on the graph, but it is not the contribution of the paper. The paper designs a way to use the Bethe Hessian operator very similar to many co-occurrence factorization methods.

While the technique is shown to produce competitive performance on word similiarity tasks, this result, unfortunately, has few practical implications as there is a plethora of word embeddings techniques that can be used to achieve the same level of performance (some easier to use than others).

Perhaps the work may benefit from focusing on the technique aspects of the approach, for instance analyzing the properties of the method. One missing reference that is closely relavant to the work is the long line of spectral word embedding methods using canonical correlation analysis: for example, see Stratos et al. (2015, 2014) and Dhillon et al. (2011).

---

### Official Review · AnonReviewer2 · 2018-03-10
**Well written and**

**Rating:** 8
**Confidence:** 4

**Review:**

In the present paper, the authors extend a spectral method for clustering  to learn dense word representations, an algorithm they name SpectralWords. They show, through extensive experiments, that their approach achieves comparable or superior to the state-of-the-art performance.

This short paper looks like is a straightforward accept yo me: the method is interesting, it uses an innovative approach, and has good performance on data sets. The paper is short, yet well written and the litterature is well covered.

---

### Decision · Program_Chairs · 2018-03-20
**ICLR 2018 Workshop Acceptance Decision**

**Decision:**

Accept

**Comment:**

Congratulations, your paper was accepted to the ICLR workshop.